# Assessing the Effects of Rotifer Feed Enrichments on Turbot (*Scophthalmus maximus*) Larvae and Post-Larvae Gut-Associated Bacterial Communities

**DOI:** 10.3390/microorganisms11020520

**Published:** 2023-02-17

**Authors:** Antonio Louvado, Carolina Castro, Davide A. M. Silva, Vanessa Oliveira, Luís E. C. Conceição, Daniel F. R. Cleary, Newton C. M. Gomes

**Affiliations:** 1Department of Biology and Centre for Environmental and Marine Studies (CESAM), University of Aveiro, 3810-193 Aveiro, Portugal; 2Flatlantic—Actividades Piscícolas, SA 3070-732 Praia de Mira, Portugal; 3SPAROS Lda., 8700-221 Olhao, Portugal

**Keywords:** live feed, larviculture, Roseobacter, microbial modulation, *Vibrio*, *Brachionus plicatilis*

## Abstract

Live feed enrichments are often used in fish larvicultures as an optimized source of essential nutrients to improve larval growth and survival. In addition to this, they may also play an important role in structuring larval-associated microbial communities and may help improve their resistance to diseases. However, there is limited information available on how larval microbial communities and larviculture water are influenced by different live feed enrichments. In the present study, we investigated the effects of two commercial rotifer enrichments (ER) on turbot (*Scophthalmus maximus*) larval and post-larval gut-associated bacterial communities during larviculture production. We evaluated their effects on bacterial populations related to known pathogens and beneficial bacteria and their potential influence on the composition of bacterioplankton communities during larval rearing. High-throughput 16S rRNA gene sequencing was used to assess the effects of different rotifer enrichments (ER1 and ER2) on the structural diversity of bacterial communities of the whole turbot larvae 10 days after hatching (DAH), the post-larval gut 30 DAH, and the larviculture water. Our results showed that different rotifer feed enrichments were associated with significant differences in bacterial composition of turbot larvae 10 DAH, but not with the composition of larval gut communities 30 DAH or bacterioplankton communities 10 and 30 DAH. However, a more in-depth taxonomic analysis showed that there were significant differences in the abundance of Vibrionales in both 10 DAH larvae and in the 30 DAH post-larval gut fed different RE diets. Interestingly, the ER1 diet had a higher relative abundance of specific amplicon sequence variants (ASVs) related to potential *Vibrio*-antagonists belonging to the Roseobacter clade (e.g., *Phaeobacter* and *Ruegeria* at 10 DAH and *Sulfitobacter* at 30 DAH). In line with this, the diet was also associated with a lower relative abundance of *Vibrio* and a lower mortality. These results suggest that rotifer diets can affect colonization by *Vibrio* members in the guts of post-larval turbot. Overall, this study indicates that live feed enrichments can have modulatory effects on fish bacterial communities during the early stages of development, which includes the relative abundances of pathogenic and antagonist taxa in larviculture systems.

## 1. Introduction

Wild finfish larvae feed on a diverse array of zooplankton (e.g., copepods, cladocerans, ciliates, and rotifers [1]) in order to fulfill their nutritional requirements during early life stages [2,3], with copepods of particular importance [3,4]. Copepods are naturally rich in essential nutrients (e.g., high quality proteins, amino acids, and minerals) and polyunsaturated fatty acids (PUFAs, e.g., ω-3 and ω-5 [3,5]). PUFAs are essential for larval development, and a deficiency or imbalance in PUFAs may adversely affect fish growth and immunological development [6,7,8,9]. However, the mass cultivation of these small crustaceans is challenging, and, despite some successful small-scale ventures [10], they are not systematically used in the marine finfish larviculture sector [3,11]. As an alternative, rotifer and artemia are the most commonly used as the first feed in larviculture [5,11]. However, both artemia and rotifer are naturally poor in PUFAs. To overcome this problem, a range of commercial products, made of nutritionally balanced lipid emulsions, have been approved for the enrichment of live feed with PUFAs and are commonly used in the production of fish larvae [5,9]. An optimal nutrition of these commercial enrichments has been intensively debated in the literature for most commercial finfish species, including the turbot (*Scophthalmus maximus*) [12], and this has led to significant improvements in fish survival, growth, and disease resistance [13,14].

In addition to the importance of providing nutritionally complete live diets, the continuous production of marine finfish fingerlings is frequently affected by stochastic episodes of mass mortalities in the first weeks of life [2,15]. These outbreaks have typically been associated with pathogens belonging to genera *Vibrio*, *Photobacterium*, and *Tenacibaculum* that naturally inhabit the microbial communities of rotifers, copepods, and artemia [16,17,18]. To tackle this problem, strategies based on the exposure of live feed and fish eggs to mild doses of chemical disinfectants have been adopted as the standard procedure in the larviculture industry [19]. Although provisionally effective, these strategies may interfere with the normal development of fish larvae [20] and the quality of the live feed [21]. Furthermore, the process of disinfection affects the microbial community as a whole, creating a more unstable environment, which will favor the development of fast-growing opportunistic r-strategists and increase the probability of pathogen proliferation and fish larvae infection [22]. Despite the perception that the disinfection processes can have significantly adverse effects on the larviculture microbiome, there is a scarce amount of information on how live feed enrichments affect the initial microbe recruitment of newly hatched larvae [15,23,24,25].

The establishment of larval microbial communities is a multifactorial process, driven by both extrinsic (e.g., environment) and intrinsic processes (e.g., phylogeny and host genetics; [26,27,28]), in which diet plays a crucial role [25,29,30]. In larviculture, the rearing water and feed are the earliest sources of bacteria that colonize the skin and gastrointestinal tracts of fish [15,24,29,31]. There is, however, a lack of information on how larval bacterial communities are influenced by different commercial live feed enrichments. Therefore, in line with the need to better understand the effects of live diets on larval microbiomes, in the present study we aimed to investigate the effects of two commercial rotifer enrichments (ER1 and ER2) on turbot (*Scophthalmus maximus*) larval and post-larval gut-associated bacterial communities during larviculture production. We hypothesize the following: (1) Rotifer feed enrichments will have distinct effects on turbot (*Scophthalmus maximus*) larvae and post-larval gut-associated bacterial communities. (2) Different live feed enrichment products will have different effects on the balance of potential pathogenic and antagonistic microbes in the larviculture production system. (3) Bacterial populations of live feed enrichments will colonize turbot larvae and post-larval gut biotopes.

## 2. Methods

### 2.1. Larviculture

Water, larvae, post-larvae, and live feed samples were obtained between 17 November and the 14 December 2021 in an intensive larviculture facility for the production of turbot (*S. maximus*) larvae located in Portugal. The larviculture operates in a flow-through system with natural seawater collected from the adjacent Atlantic Ocean. Sampling was conducted during routine and animal husbandry practices of commercial farming operations. The sampling procedure was in line with approved methods of animal killing according to Annex IV of the European Directive Dir2010/63/EU on the protection of animals used for scientific purposes. No experimental procedure was conducted on live fish. Two products (ER1 and ER2), commercialized as enrichment diets for live feed, were used as rotifer enrichments. Both products are commercially available and have been previously approved for use for animal feeding. They are routinely used by the larviculture facility where the sampling occurred. ER1 and ER2 are dry formulas enriched in polyunsaturated fatty acids (PUFAs), vitamins, and minerals. Based on the content declarations, ER1 contains a taurine additive and has higher protein and PUFA content than ER2. ER2 has a higher vitamin content than ER1 and a probiotic additive based on viable cells of a strain of *Pediococcus acidilactici*. Due to the fact that this study was not designed to evaluate different commercial live feed enrichments, but instead to investigate their importance in shaping larval microbiomes, the commercial name of the products is not provided, in order to avoid misjudgment of the efficiency and quality of different products.

Water and larvae samples were obtained from independent cylindrical tanks (three for each live feed enrichment). Larviculture tanks were operated as a flow-through system with water pH 7.6–8.0, temperature 19.18 ± 0.51 °C, and salinity 34.05 ± 0.11‰. Larvae were fed rotifers, enriched by ER1 or ER2, from 3 to 15 DAH. Rotifers enriched with their respective product (ER1 or ER2) were added to rearing water of their respective tanks, together with a mature culture of *Tetraselmis* sp. (green water technique). From 11 to 15 DAH, the differently enriched rotifers were gradually replaced by artemia enriched with ER2. Between 16 DAH and until 30 DAH, only artemia enriched with ER2 were provided. Fish were fed manually with a three hour interval between feeds. At 30 DAH, fish were transferred to nursery tanks and they remained there until 60 DAH. During this time, any detected abnormal fish (i.e., small, deformed, diseased, or depigmented fish) were euthanized. At 60 DAH, surviving fish were counted to determine the survival rate of each tank.

### 2.2. Sample Collection and DNA Extraction

Composite samples of the whole turbot larvae (n = 25) and the post-larval gut (n = 5) were obtained seven (10 DAH) and twenty-seven days (30 DAH) after their transfer to the larval rearing system, respectively. One composite sample was obtained for each tank, totaling three composite samples of larvae for each combination of treatments (ER1 and ER2) and sampling points (10 and 30 DAH). Larvae were collected using a fish net (the fish net was cleaned in-between samplings with a 0.1% hydrogen peroxide solution). Collected larvae were transferred to 120 mL sterile plastic cups filled with respective rearing water. Larvae were killed using an overdose of fish anesthetic (clove oil; Sigma-Aldrich, Saint Louis, MO, USA) and then rinsed with sterile artificial seawater with salinity adjusted to 36 ppt (ASW). Dissection only occurred after proof of death (waiting for the onset of rigor mortis). Smaller larvae (10 DAH) were rinsed five times with 5 mL of ASW in a 15 mL falcon tube, while larger larvae (30 DAH) were washed on both sides for approximately 5 s with ASW, using a previously autoclaved wash bottle. Twenty-five larvae with 10 DAH, were transferred whole into bead-beating tubed from the DNA extraction kit. Five larvae with 30 DAH, were dissected and their gastro-intestinal tract was retrieved whole and pooled into bead-beating tubes from the DNA extraction kit. 

Bacterioplankton samples consisted of rearing water collected from each tank at the same time that larvae samples were collected (i.e., 10 and 30 DAH). A 250 mL sterile Erlenmeyer flask was filled with rearing water and filtered through a 0.22 µm polycarbonate filter (Merck-Millipore, Rahway, NJ, USA) in an EZ-Fit™ Manifold base (Merck-Millipore, Rahway, NJ, USA) or in a Büchner filtration assembly connected to a vacuum pump. After filtration, the polycarbonate filter was cut thoroughly using sterile scissors and was transferred to a bead-beating tube. Live feeds (rotifers, algae, and artemia) were collected from respective stock cultures using a sterile 250 mL cup and transported to the laboratory. Only one sample of each stock culture of feed was collected in the experiment. Algae and rotifer samples were collected seven days after larval transfer to the rearing system, while artemia samples were collected twenty-seven days later. Algae were harvested by centrifugation at 4400× *g* during 30 min; supernatant was discarded and the pellet was suspended in DNA extraction buffer and transferred into bead-beating tubes from the DNA extraction kit. Rotifer and artemia samples were collected by filtering samples through a Whatman 113 paper filter (Merck-Millipore, Rahway, NJ, USA). Residues were washed three times with sterile artificial seawater, collected using a sterile spatula (300 mg), and transferred into respective bead-beating tubes from the DNA extraction kit.

All samples (larvae, bacterioplankton, and feed) were processed using FastDNA^TM^ Spin kit (MP Biomedicals, Santa Ana, CA, USA) following the manufacturer’s instructions. A blank negative control with no sample was included in the DNA extraction process each time. All samples were processed immediately; no storage was required.

### 2.3. High-Throughput Sequencing Data Acquisition

The hypervariable region V3/V4 of the 16S rRNA gene was amplified by PCR using the primers 314F (CCTACGGGNGGCWGCAG) and 785R (GACTACHVGGGTATCTAATCC) [32]. Library preparation and sequencing were performed using a MiSeq sequencing platform at Molecular Research LP (www.mrdnalab.com; accessed on 15 September 2022, Shallowater, TX, USA), following standard Illumina procedures (Illumina, San Diego, CA, USA). QIIME2 (version 2020.8) was used to transform the amplicon libraries to an amplicon sequence variant (ASV) abundance table [33]. Demultiplexing was performed using the “demux” algorithm in QIIME2. The dada2 algorithm from the DADA2 plugin [34] in QIIME2 was used to filter low-quality reads, merge forward and reverse reads into sequences, remove chimeras, and group sequences into ASVs. In dada2, forward and reverse sequences were truncated at 220 and 240 bp, respectively. Taxonomy was assigned to ASVs using the ‘feature-classifier’ algorithm in QIIME2 with a scikit-learn Naïve Bayes classifier based on the SILVA database of the 16S reference sequences at 99% similarity (version 138, released December 2019). The classifier was previously trained using a ‘feature-classifier’ algorithm in QIIME2 (version 2020.8) with reference sequences, trimmed and truncated at the 314F and 785R region. To simplify interpretation, a unique number was assigned to each ASV. Non-bacterial, mitochondrial, and chloroplastidial sequences were removed. Negative controls from the DNA extraction were also sequenced and ASVs that occurred in the negative controls, but which did not appear to be the result of “index hopping” [35], were removed. A list of removed ASVs is presented in Appendix A.

### 2.4. Data Analysis and Statistics

A table containing the ASV counts per sample was imported into R and used to compare community diversity and composition and assess the relative abundance of selected higher taxa. The Shannon’s H’ diversity index was calculated using the diversity() function in the vegan package [36], rarefied richness was calculated using the rarefy() function in vegan, evenness (Pielou’s J) was calculated by dividing Shannon H’ by the logarithm of the total number of ASVs, and Fisher’s alpha was calculated using the fisher.alpha() function in vegan. Since diversity parameter data were not normally distributed, we tested for significant variations for each factor separately: diet, biotope (water and larvae), and sampling time; using the Kruskal–Wallis rank sum test with the kruskal.test() function in the stats package. Relative abundances of selected bacterial higher taxa were tested for significant variation between diets, biotopes (water and larvae), and sampling times using the anova() function in R with the F test applied to a general-linearized model using the glm() function in R [37]. Since a number of these variables included an excess of zero counts in the samples, we set the family argument to ‘tweedie’ using the tweedie() function in the statmod package in R, with var.power = 1.5 and link.power = 0 (a compound Poisson–gamma distribution). To perform pairwise comparisons among treatments, we used the emmeans() function from the emmeans package with the p-value adjustment set to the false discovery rate (i.e., p.adjust = ”fdr”). Biplot ordinations of ASV composition were produced for the full dataset and separately for each sampling point (10 and 30 DAH). First, a phyloseq object was generated using the phyloseq() function from the ‘phyloseq’ package [38]. The ordinate() function in ‘phyloseq’ was subsequently used with the phyloseq object as input, the method argument set to ‘PCoA’, and the distance argument set to ‘bray’. A biplot was then produced using the plot_ordination() function in ‘phyloseq’ with the type argument set to ‘biplot’. Significant differences among factors (diet (ER1 vs. ER2) and sampling point (10 vs. 30 DAH) and biotope (water vs. larvae)) and their interactions were determined using the adonis() function in vegan for a permutational multivariate analysis of variance (PERMANOVA). The BLAST search tool (http://www.ncbi.nlm.nih.gov/; accessed on 15 September 2022) was used to compare representative sequences of the 50 most abundant ASVs to sequences in the NCBI 16S ribosomal RNA (Bacteria and Archaea type strains) database using standard parameter settings [39,40]. Sequences that exhibited the highest levels of similarity were considered to be closely related organisms.

## 3. Results and Discussion

After quality control and the removal of non-bacterial sequences, the dataset consisted of 2486757 sequences binned into 2336 ASVs. Overall, our results showed similar α-diversity patterns for water and fish (larvae 10 DAH and post-larval gut 30 DAH) bacterial communities in tanks with different rotifer diets (ER1 and ER2) (Figure 1; Kruskal Wallis–Wallis: *p* > 0.05). This is in line with other studies, which have shown that live feed manipulation did not affect rarefied richness of the gut-associated bacterial communities of 34 DAH gilthead seabream post-larvae (*Sparus aurata*; [15]). Bacterioplankton communities, however, were richer (Kruskal–Wallis: H(1) = 17.38; *p* < 0.001) than the fish samples, but there were no significant differences in evenness or Shannon’s *H’* diversity. Previous studies also showed that bacterioplankton communities tended to be richer than the host-associated bacterial communities [41,42]. There were differences in the diversity and evenness of bacterioplankton communities from different sampling events (10 vs. 30 DAH), but these differences were not significant following the *p*-value correction using the FDR method (Kruskal–Wallis: FDR-P > 0.05).

A PCO ordination highlighting the differences in ASV composition is presented in Figure 2. Biotope (water vs. larvae) and sampling time (10 vs. 30 DAH) were better predictors of the bacterial community structure than the rotifer’s diet (Figure 2). The rotifer’s diet, however, did have a significant independent and interaction effect on the ASV composition (PERMANOVA: *p* < 0.05, Table 1). Separate PCO ordinations for 10 and 30 DAH samples are presented in Appendix A in order to better evaluate the interactions between the sampling time and rotifer diets. Our results showed that the differences between different diets (ER1 and ER2) were more prevalent for the 10 DAH whole larvae samples as opposed to the post-larval gut and water samples, showing a clear separation of the effects of the different diets along the second axis. In line with this, the results of the PERMANOVA showed that the rotifer’s diet was a significant predictor of ASV composition at 10 DAH (PERMANOVA: F_1,11_ = 5.21; *p* = 0.002), but not at 30 DAH (PERMANOVA: F_1,11_ = 1.324; *p* = 0.202). Previous studies have reported similar results in cod larvae that suggested that the environmental setting (e.g., water and biofilm) is more influential than live feed during initial microbiome recruitment in larval guts [29]. Some speculate that the maladaptation of feed’s microbes to the gut environment impedes a successful gut colonization [24].

The taxonomic composition of the dataset is presented in Figure 3. Overall, our results showed that, in contrast to the fish larvae 10 DAH, the taxonomic abundances of the bacterial communities inhabiting the post-larval gut 30 DAH were highly variable among the replicates. Larval microbiomes changed rapidly during development, a process which may result in varying degrees of resistance (open niches and/or lack of antagonistic traits) to the invasion by fast-growing opportunistic microbes [43]. Our results also showed that the bacterial communities of the larvae 10 DAH and the post-larval gut 30 DAH were dominated by members of the order Vibrionales. The predominance of Vibrionales during larval development is in line with other studies on turbot larvae [44,45]. Although not all Vibrionales strains cause infection in fish, some members of this order (e.g., *Photobacterium damselae*, *V. parahaemolyticus*, *V. anguillarum*, and *V. harveyi*) are responsible for mass mortalities in aquaculture systems [46,47]. Invertebrates are recognized as natural reservoirs of *Vibrio* and, in enclosed aquaculture facilities, their administration as live feed is considered one of the main causes of *Vibrio* infections in marine larvicultures [16,31,48]. This is why many studies have investigated the potential of probiotics, prebiotics, postbiotics, and other microbiome-modulating strategies to control *Vibrio* populations in live feed [15,16,44,49,50]. Here, we also observed that the order Rhodobacterales showed a stronger dominance in the bacterial communities of the post-larval gut 30 DAH than in the whole larvae 10 DAH. This enrichment was, furthermore, more pronounced for fish larvae fed rotifers enriched with ER1 than with ER2. Overall, the majority of sequences assigned to the Rhodobacterales order were assigned to members of the *Roseobacter* clade (Appendix A). The *Roseobacter* clade is a paraphyletic group belonging to the Rhodobacterales order, with several members known to possess antagonistic activity against *Vibrio* spp. and other aquaculture pathogens, e.g., *Phaeobacter inhibens*, *P. gallaeciensis*, *P. piscinae*, *Ruegeria* sp., and *Sulfitobacter* sp. [51,52,53].

In this study, the bacterioplankton was dominated by the orders Flavobacteriales and Rhodobacterales at both sampling points (10 and 30 DAH), with a high abundance of order Oceanospirillales at 10 DAH and Pseudomonadales at 30 DAH (Figure 3). All orders except Pseudomonadales have been previously found to be abundant in the rearing water of RAS for adult turbot [54]. The most abundant ASVs of order Flavobacteriales were most similar (>98%) to sequences in the NCBI database identified as belonging to genera *Polaribacter*, *Winogradskyella*, or *Meridianimaribacter* (Appendix A). All these genera are commonly found in marine settings, including aquaculture systems [54,55,56,57]. The most abundant ASVs assigned to the Oceanospirillales order were assigned to family Nitrincolaceae and had very low similarity (<93%) with type strains of the NCBI database. Members of the Nitrincolaceae family are heterotrophic bacteria usually found as core members of bacterioplankton communities and involved in nitrogen and carbon cycling following phytoplankton blooms [58,59]. The enrichment of Pseudomonadales in the rearing water 30 DAH was mainly due to the enrichment of two ASVs (ASVs 10 and 13; Appendix A) assigned to *Paraperlucidibaca baekdonensis* (Appendix A). This specie was first isolated from marine waters [60], but was later found to be a core member of the bacterioplankton of marine aquacultures. However, its potential functional roles in aquaculture systems remain unknown [61,62].

A more in-depth pairwise analysis preformed on the six most abundant orders in the dataset showed that larvae 10 DAH had a significant and higher abundance of Vibrionales than all the other groups analyzed (Appendix A, EMMEANS: *p* < 0.05; Figure 4). Our results also showed that the abundance of this order was significantly higher in fish fed rotifers enriched with ER2 (88.384 ± 1.728% in larvae 10 DAH and 55. 919 ± 20.682% in the post-larval gut 30 DAH) than with ER1 (70.52 ± 1.187% in larvae 10 DAH and 22.033 ± 17.74% in the post-larval gut 30 DAH). Among the 50 most abundant ASVs, all those assigned to order Vibrionales were also assigned to the *Vibrio* genus (Appendix A). Interestingly, a Venn diagram analysis showed that about 92 ASVs detected in rotifers (combined ER1 and ER2) were also detected in the bacterial communities of the post-larval gut and/or in the bacterioplankton 30 DAH, algae, and artemia (Appendix A). We also observed that 51 of these ASVs were either present in fish guts’ bacterial communities alone or in both the fish guts’ bacterial communities and the bacterioplankton. This suggests that these ASVs may have a more specific association with the turbot post-larval gut. The relative abundance analysis showed that the majority of these ASVs belonged to the rare biosphere and only sixteen ASVs had relative abundances exceeding 0.1% (Appendix A). Ten of these sixteen ASVs were classified to genus *Vibrio* (ASV 51, 52, 53, 56, 57, 61, 62, 65, 69, and 76) and the remaining to genera *Litoribacillus* (ASV 59), *Peptoniphilus* (ASV 68), *Paraperlucidibaca* (ASV 10), *Conchiformibius* (ASV 75), and *Thalassolituus* (ASV 81). In general, members of *Litoribacillus*, *Paraperlucidibaca*, and *Thalassolituus* are often associated with marine bacterioplankton [60,63,64], while *Conchiformibius* and *Peptoniphilus* were previously found to be associated with oral cavities [65] and the gut of animal hosts [66,67], respectively. ASV-51 (*Vibrio* sp.) was the only ASV to have a relatively high abundance in both diets (ER1 and ER2). In general, ASVs related to *Vibrio* were less abundant in the bacterial communities of the post-larval gut fed rotifers enriched with ER1. Our results suggest that rotifer diets contribute to colonization by *Vibrio* members in the gut of turbot post-larvae, and the degree of colonization depends on the type of enrichment used.

The *Vibrio* genus is known to comprise both non-pathogenic and pathogenic strains, with the latter being a major concern for the aquaculture sector [46,47]. Interestingly, in line with the reduction in the relative abundance of *Vibrio* in ER1 samples, we also observed that this diet increased the relative abundance of ASVs assigned to the genera *Ruegeria* (ASV 47) and *Nautella* (ASV 49) at 10 DAH and *Sulfitobacter* at 30 DAH (ASV-2; Figure 5 and Appendix A). Nonetheless, sequence similarity analysis using the NCBI type strain database showed that ASV 49 had the highest similarity (100%) with a strain previously identified as *Nautella italica*, but later reclassified as *Phaeobacter italicus* (Appendix A; [68]). *Ruegeria*, *Phaeobacter*, and *Sulfitobacter* are bacterial members of the *Roseobacter* clade [69] and are known to comprise several bacterial strains with antagonistic activity against bacterial pathogens, including *Vibrio* species [52,70]. Interestingly, the lower *Vibrio* abundance and higher abundance of potential antagonistic bacteria (*Ruegeria*, *Phaeobacter* and *Sulfitobacter*) detected in ER1 tanks was associated with a significantly higher fish survival rate (8.0 ± 0.9%) in comparison to ER2 (4.34 ± 0.89%). The survival rates reported here at 60 DAH are similar to previous studies of turbot larviculture [14,71]. It has been recognized that the survival of the turbot larvae is often lower than that of other marine fish [72], but in the past decades, larval survival has been a major undertaking for hatchery management, with significant advances obtained through improvements in the nutritional formulation of live feed enrichment [13,14].

## 4. Conclusions

This study showed that different rotifer enrichments were associated with significant differences in the bacterial community structure of turbot larvae 10 DAH but not with post-larval gut communities 30 DAH. However, a more in-depth analysis showed that the ER1 diet was associated with a lower abundance of *Vibrio* in whole larvae and post-larval gut-associated communities, and a higher abundance of the ASVs related to potential *Vibrio* antagonists (i.e., members of the Roseobacter clade). Overall, our findings indicate that live feed enrichment formulas can have modulatory effects on fish communities during their early stages of development with the potential to affect the balance of pathogenic and antagonist microbes in larviculture. In addition to the above, different enrichments potentially contribute to *Vibrio* colonization in the guts of post-larval turbot; the degree of colonization may also vary depending on the enrichment used. Our findings highlight the importance of analyzing the effects of differentially enriched rotifer feeds on the structural composition of microbial communities in fish larviculture. Given the uniqueness of microbial communities at different larviculture systems and the well-known effects of the host microbiome on animal health and growth [73], aquaculture producers should consider the strategy applied in the present study to evaluate and optimize larval fish diets.

## Figures and Tables

**Figure 1 microorganisms-11-00520-f001:**
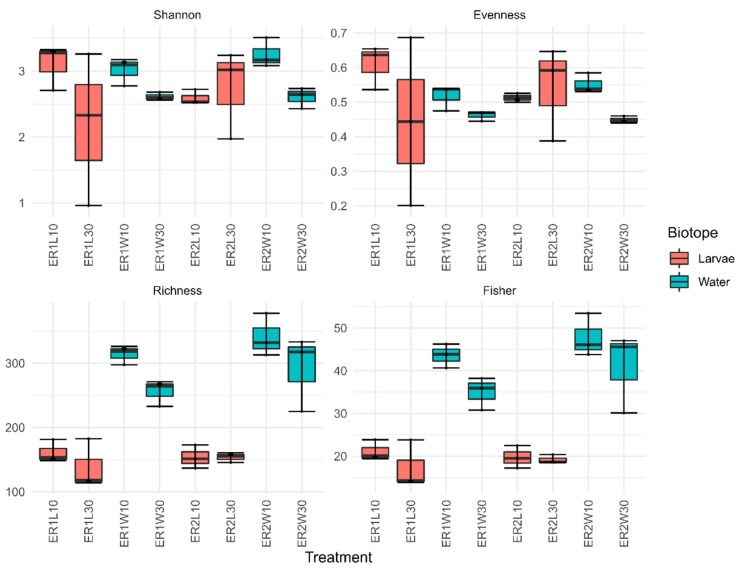
Diversity indices (Peilou’s J (evenness), richness, Shannon’s H’ (Shannon), and Fisher’s α (Fisher)) of bacterial communities of larvae (10 DAH), post-larval gut (30 DAH) and respective water samples, in tanks fed rotifer enriched with different commercial products (ER1 and ER2). ER1L10 and ER2L10 (whole larvae 10 DAH); ER1L30 and ER2L30 (post-larval gut 30 DAH); ER1W10 and ER2W10 (rearing water 10 DAH); and ER1W30 and ER2W30 (rearing water 30 DAH).

**Figure 2 microorganisms-11-00520-f002:**
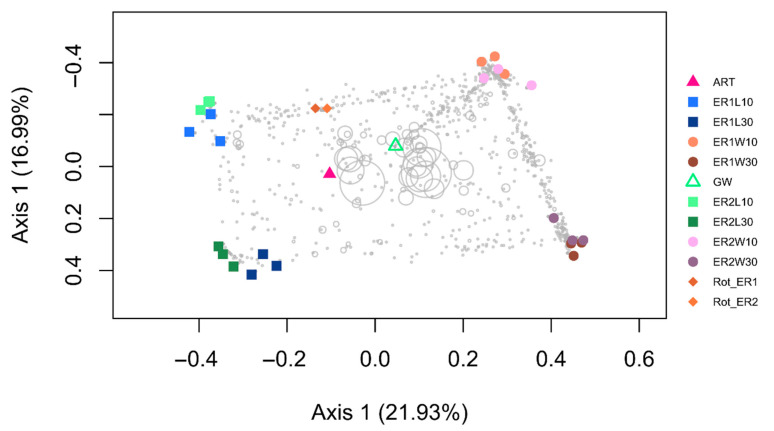
First two axes of a principal coordinates analysis (PCO) of ASV composition. Colored symbols are samples. Gray symbols are weighted averages scores for ASVs (size is proportional to abundance). ER1 and ER2 corresponds to the different rotifer diets. ART (Artemia 30 DAH); GW (Algae 10 DAH); Rot_ER1 and Rot_ER2 (Rotifers 10 DAH); ER1L10 and ER2L10 (whole larvae 10 DAH); ER1L30 and ER2L30 (post-larval gut 30 DAH); ER1W10 and ER2W10 (rearing water 10 DAH); and ER1W30 and ER2W30 (rearing water 30 DAH).

**Figure 3 microorganisms-11-00520-f003:**
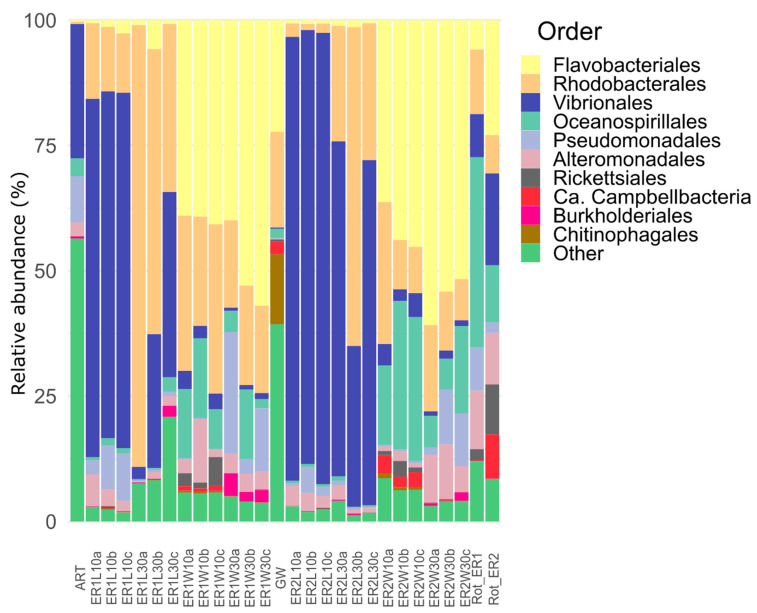
Taxonomic composition at order level for feed, larval, and water samples in tanks fed rotifers enriched with different commercial products (ER1 and ER2) at 10 and 30 DAH. ART (artemia); GW (green water (i.e., mature culture of *Tetraselmis* sp.)); Rot_ER1 and Rot_ER2 (rotifer); ER1L10 and ER2L10 (whole larvae 10 DAH); ER1L30 and ER2L30 (post-larval gut 30 DAH); ER1W10 and ER2W10 (rearing water 10DAH); and ER1W30 and ER2W30 (rearing water 30 DAH). Small letters indicate tank-replicates. Data are present as relative abundances of the ten most abundant orders for individual tank-replicates.

**Figure 4 microorganisms-11-00520-f004:**
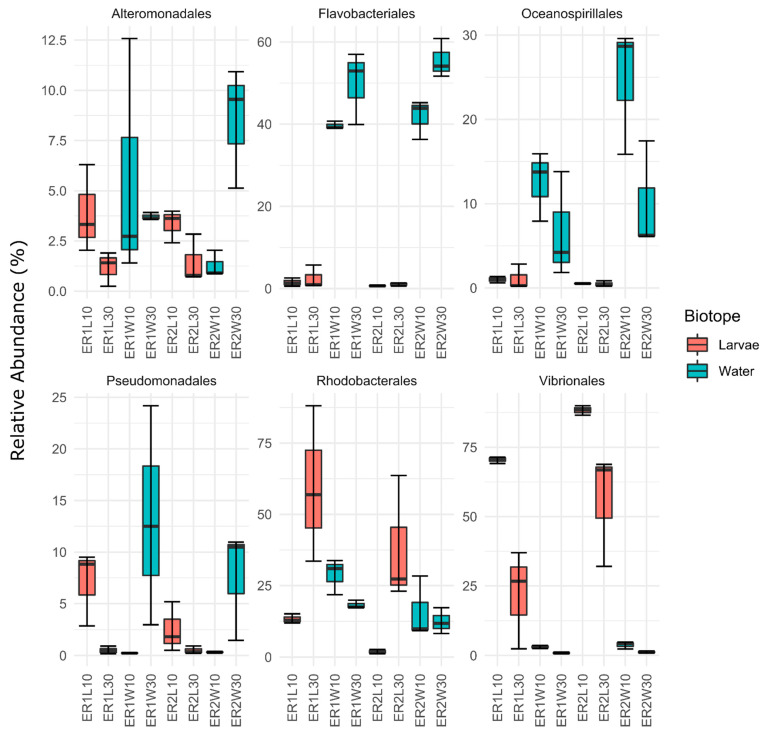
Relative abundances of the six most abundant bacterial orders for larval and water samples in rotifer-enriched feeds with different commercial products (ER1 and ER2). ER1L10 and ER2L10 (whole larvae 10 DAH); ER1L30 and ER2L30 (post-larval gut 30 DAH); ER1W10 and ER2W10 (rearing water 10 DAH); and ER1W30 and ER2W30 (rearing water 30 DAH).

**Figure 5 microorganisms-11-00520-f005:**
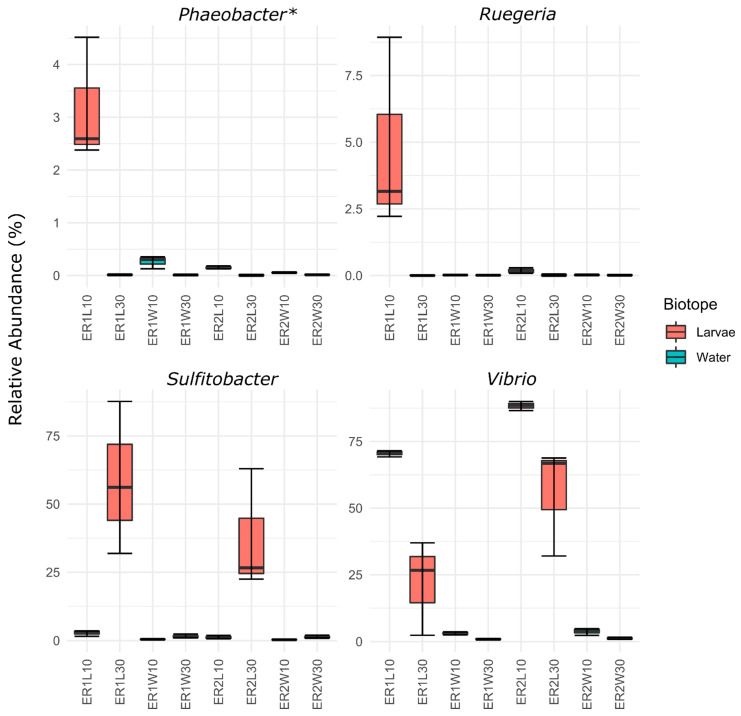
Relative abundance of ASVs assigned to the genera *Phaeobacter*, *Sulfitobacter*, *Ruegeria*, and *Vibrio* in rotifer-enriched feeds with different commercial products (ER1 and ER2). ER1L10 and ER2L10 (whole larvae 10 DAH); ER1L30 and ER2L30 (post-larval gut 30 DAH); ER1W10 and ER2W10 (rearing water 10 DAH); and ER1W30 and ER2W30 (rearing water 30 DAH). * Genus *Phaeobacter* includes all ASVs classified as *Phaeobacter* by the naïve Bayes classifier based using the SILVA database of the 16S reference sequences at 99% similarity (version 138, released December 2019) and all ASVs classified as *Nautella* but with high similarity (>99%) to type strains of the genus *Phaeobacter* in sequence similarity search of the NCBI 16S database using the BLAST algorithm.

**Table 1 microorganisms-11-00520-t001:** PERMANOVA results of variation in ASV composition among groups and their interactions using the adonis() function in vegan applied to the Bray–Curtis dissimilarity matrix of a log-transformed ASV table and with permutations set to 999. Factors tested include rotifer’s diet (Diet; ER1 and ER2), Biotope (water and larvae), and fish age (Age; 10 and 30 DAH). *p*-values (P) were adjusted by false-discovery rate (FDR-P). Significant values (Sig.) are presented as **—FDR-P < 0.01, *—FDR-P < 0.05, and ns—not significant.

Factor	F_1,23_	R^2^	P	FDR-P	Sig.
Diet	2.4999	0.02844	0.037	0.432	*
Biotope	25.7843	0.29331	0.001	0.002	**
Age	19.0044	0.21619	0.001	0.002	**
Diet:Biotope	2.0005	0.02276	0.064	0.064	ns
Diet:Age	3.0973	0.03523	0.010	0.015	*
Biotope:age	16.5838	0.18865	0.001	0.002	**
Diet:Biotope:Age	2.9368	0.03341	0.011	0.015	*

## Data Availability

Sequences used in this study were uploaded to the NCBI ShortRead Archive (BioProject PRJNA911963; Biosamples SAMN32217043—SAMN32217070).

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
