# Peer review of "Assessing the Effects of Rotifer Feed Enrichments on Turbot (Scophthalmus maximus) Larvae and Post-Larvae Gut-Associated Bacterial Communities"

_microorganisms, 2023, doi:10.3390/microorganisms11020520_

Round 1

Reviewer 1 Report

The study aimed at understanding the effect of live feed on bacterial communities in larviculture. The study is interesting and well thought through. 

General Comments:

1.      What can be done to avoid the potential negative impact of rotifers on bacterial communities?

2.      The full name of ASVs should be provided at the first mention, preferably in each section.

3.      This study was done in a controlled environment. Are there references to similar studies in an uncontrolled environment? If so what can we learn from those studies? Whether extrinsic factors such as temperature variation can change the result in your case.

4.      The second paragraph of the introduction is too long. The authors can consider breaking it into two.

5.      Line 78 …understand of the effects of…. Should be …understand the effects of …

6.      Line 84 …products have specific effects on … should be …products have different effects on …

7.      Line 89 …Samples were obtained between… Please specify the samples.

8.      Line 109-111: The authors should include where the experiment was carried out.

9.      Line 111 -112: Indicate the feeding rate.

10.   113-116: Difficult to comprehend. Could the authors rewrite this in much clearer words? It should be clear what the biological groups were fed with, and which groups were replaced with what.

11.   Line 159-161: Include the source of the sequence.

12.   Line 317: Supplementary File S3, Should be File S3.

13.   Line 381: …modulatory effects of fish communities… Should be …modulatory effects on fish communities…

14.   The material and method should be much clearer, it should be written in a clear and orderly manner. Also detailed information on how, where the

15.   The authors should consider giving an appropriate heading to supplementary files 1 and 2.

16.   Supplementary File S1 should be Supplementary File 1 OR File S1. This should be done for other Supplementary Files, Figures, and Tables in the supplementary and main text.

17.   In File S2, Supplementary Figure S - Heatmap analysis … Should be Figure S3 - Heatmap analysis.

18.   Should the scientific names in the bibliography not be italicized? 

Author Response

Reviewer 1:

The study aimed at understanding the effect of live feed on bacterial communities in larviculture. The study is interesting and well thought through.

We wish to thank the reviewer for dispending his/her time to review the article.

General Comments:

  1. What can be done to avoid the potential negative impact of rotifers on bacterial communities?

As pointed out in the introduction section, common larviculture practices rely on antimicrobial strategies to reduce the load of bacteria in rotifer, see Lines 20-25. As an alternative we propose the modulation of these bacterial communities via their main environmental routes (water and feed), see Lines 28-32.

  1. The full name of ASVs should be provided at the first mention, preferably in each section.

The full name of ASV was provided at first mention in Abstract and in the Manuscript.

  1. This study was done in a controlled environment. Are there references to similar studies in an uncontrolled environment? If so what can we learn from those studies? Whether extrinsic factors such as temperature variation can change the result in your case.

To the best of our knowledge, no study has yet evaluated the effect different live feed enrichment products have on fish larvae microbiome in an uncontrolled environment (i.e., an extensive or semi-intensive larviculture facility). In regard to extrinsic factors (e.g., temperature), we consider that these may have an effect on initial larval microbiome, but it is not the objective of the study to evaluate them.

  1. The second paragraph of the introduction is too long. The authors can consider breaking it into two.

We have corrected this.

  1. Line 78 …understand of the effects of…. Should be …understand the effects of …

We have corrected this.

  1. Line 84 …products have specific effects on … should be …products have different effects on …

We have corrected this.

  1. Line 89 …Samples were obtained between… Please specify the samples.

We have corrected this.

  1. Line 109-111: The authors should include where the experiment was carried out.

Currently several companies demand non-disclosure agreements to allow microbiome studies in their facilities. Unfortunately, we cannot disclose the exact location and company name where the samples were collected. However, the geographical area is provided in the manuscript.

  1. Line 111 -112: Indicate the feeding rate.

We have corrected this. See line 67-68 of the revise manuscript: “Fish were fed manually with a three hour interval between feeds.”

  1. 113-116: Difficult to comprehend. Could the authors rewrite this in much clearer words? It should be clear what the biological groups were fed with, and which groups were replaced with what.

We have corrected this, highlighting that only rotifer were enriched with different ERs.

  1. Line 159-161: Include the source of the sequence.

Citation of the source of primers has been included.

  1. Line 317: Supplementary File S3, Should be File S3.

We have corrected in line with following comments.

  1. Line 381: …modulatory effects of fish communities… Should be …modulatory effects on fish communities…

We have corrected this.

  1. The material and method should be much clearer, it should be written in a clear and orderly manner. Also detailed information on how, where the

We have corrected the materials and methods section in line with previous comments. We also revised the whole document.

  1. The authors should consider giving an appropriate heading to supplementary files 1 and 2.

We have included an appropriate heading to Supplementary File S1. Supplementary File S2 already had a heading included.

  1. Supplementary File S1 should be Supplementary File 1 OR File S1. This should be done for other Supplementary Files, Figures, and Tables in the supplementary and main text.

We have corrected this throughout the manuscript and supplementary files.

  1. In File S2, Supplementary Figure S - Heatmap analysis … Should be Figure S3 - Heatmap analysis.

We have corrected this.

  1. Should the scientific names in the bibliography not be italicized?

We have updated the bibliography style to the reference style of the Microorganisms journal on Mendeley.

Reviewer 2 Report

The topic of research is of great importance in aquaculture sector.

The introduction is well-written and provides sufficient background. The materials and methods are informative and clearly presented.

Appropriate research design is also presented. 

The main question of the recent manuscript was to evaluate the use of two different enriched-rotifer feeds on the health status of turbot larvae through studying their effect on gut-associated microbes.  Which I found interesting and relevant to readers and people working in aquaculture sector. 

The topic shows a considerable degree of originality and the results add to the area of life-feed enrichment in aquatic organism feed.   

Overall, the paper is well-written and presented in clear and sound scientific manner. 

Both results and discussion are clear and cover all aspects of the experimental work.  

The conclusion is consistent with the evidence and arguments presented in the study hypothesis and clearly answered the main question posted.

Author Response

Reviewer 2:

The topic of research is of great importance in aquaculture sector.

The introduction is well-written and provides sufficient background. The materials and methods are informative and clearly presented.

Appropriate research design is also presented. 

The main question of the recent manuscript was to evaluate the use of two different enriched-rotifer feeds on the health status of turbot larvae through studying their effect on gut-associated microbes.  Which I found interesting and relevant to readers and people working in aquaculture sector. 

The topic shows a considerable degree of originality and the results add to the area of life-feed enrichment in aquatic organism feed.   

Overall, the paper is well-written and presented in clear and sound scientific manner. 

Both results and discussion are clear and cover all aspects of the experimental work.  

The conclusion is consistent with the evidence and arguments presented in the study hypothesis and clearly answered the main question posted.

We wish to thank the reviewer for dispending his/her time to review the article.

Reviewer 3 Report

In this paper, the authors well described the effects exerted by different rotifer feed enrichments on bacterial communities of turbot larvae 10 DAH, as well as the possible probiotic effect of the RE1 diet against Vibrio members. This observation was also associated with lower mortality of larvae. The results are important and have a novelty in the continuously growing awareness of the efficacy of bio alternatives to improve fish larvae survival and quality in aquaculture

    With respect to the previous literature, where other authors only evaluated the growth rate and/or survival of larvae, the additive concentration in their feed and the fatty acid composition of the animal, in the present work authors added, for the first time, the microbiota composition shifting following the dietary pattern.

    the conclusion consistent with the evidence and arguments are presented? the authors fully addressed the main question proposed in the paper and also expressed in the title

I strongly recommend the publication of the manuscript, provided that the authors revise the references style.

Author Response

Reviewer 3:

In this paper, the authors well described the effects exerted by different rotifer feed enrichments on bacterial communities of turbot larvae 10 DAH, as well as the possible probiotic effect of the RE1 diet against Vibrio members. This observation was also associated with lower mortality of larvae. The results are important and have a novelty in the continuously growing awareness of the efficacy of bio alternatives to improve fish larvae survival and quality in aquaculture

With respect to the previous literature, where other authors only evaluated the growth rate and/or survival of larvae, the additive concentration in their feed and the fatty acid composition of the animal, in the present work authors added, for the first time, the microbiota composition shifting following the dietary pattern.

the conclusion consistent with the evidence and arguments are presented? the authors fully addressed the main question proposed in the paper and also expressed in the title

I strongly recommend the publication of the manuscript, provided that the authors revise the references style.

We wish to thank the reviewer for dispending his/her time to review the article. We have updated the reference style of the manuscript to meet the requirements of the Microorganisms journal.